# Searching for Peripheral Biomarkers in Neurodegenerative Diseases: The Tryptophan-Kynurenine Metabolic Pathway

**DOI:** 10.3390/ijms21249338

**Published:** 2020-12-08

**Authors:** Nóra Török, Masaru Tanaka, László Vécsei

**Affiliations:** 1MTA-SZTE, Neuroscience Research Group, Semmelweis u. 6, H-6725 Szeged, Hungary; toronora85@gmail.com (N.T.); tanaka.masaru.1@med.u-szeged.hu (M.T.); 2Department of Neurology, Interdisciplinary Excellence Centre, Faculty of Medicine, University of Szeged, Semmelweis u. 6, H-6725 Szeged, Hungary

**Keywords:** neurodegenerative disease, Alzheimer’s disease, Parkinson’s disease, amyotrophic lateral sclerosis, Huntington’s disease, multiple sclerosis, tryptophan, kynurenines, biomarkers, personalized medicine

## Abstract

Neurodegenerative diseases are multifactorial, initiated by a series of the causative complex which develops into a certain clinical picture. The pathogenesis and disease course vary from patient to patient. Thus, it should be likewise to the treatment. Peripheral biomarkers are to play a central role for tailoring a personalized therapeutic plan for patients who suffered from neurodegenerative diseases such as Alzheimer’s disease, Parkinson’s disease, and multiple sclerosis, among others. Nevertheless, the use of biomarkers in clinical practice is still underappreciated and data presented in biomarker research for clinical use is still uncompelling, compared to the abundant data available for drug research and development. So is the case with kynurenines (KYNs) and the kynurenine pathway (KP) enzymes, which have been associated with a wide range of diseases including cancer, autoimmune diseases, inflammatory diseases, neurologic diseases, and psychiatric disorders. This review article discusses current knowledge of KP alterations observed in the central nervous system as well as the periphery, its involvement in pathogenesis and disease progression, and emerging evidence of roles of microbiota in the gut-brain axis, searching for practical peripheral biomarkers which ensure personalized treatment plans for neurodegenerative diseases.

## 1. Introduction

More and more physicians are considering the use of evidence-based personalized medicine, a medical model that sorts patients into different groups according to genomics, data analytics, and population health to tailor individual therapy with the best therapeutic response as well as the highest safety margin to ensure the most appropriate care. Biomarkers play an essential role for the selection of high-risk population, determination of disease subtypes, prediction of disease progression, choice of treatment, and identification of disease targets. Furthermore, biomarkers play a crucial role in drug development. Since the launch of biomarker research in the early 2000s, an increasing number of studies have been presented in scientific community [1]. However, a disproportionally low number of biomarkers are employed for clinical practice, compared to that of biomarkers documented in scientific literature. It is mostly due to low numbers of participants for biomarker research, resulting low statistical power, and a lack of the validation and standardization for methods used [2]. Thus, discovery and development of reliable peripheral biomarkers are essential for the sake of personalized medicine.

The tryptophan (TRP)-kynurenine (KYN) metabolic pathway is the main catabolic route of TRP metabolism through which over 95% of TRP degrades into several bioactive metabolites including proinflammatory, anti-inflammatory, oxidative, antioxidative, neurotoxic, neuroprotective, and/or immunologic compounds [3]. Furthermore, the KP enzymes directly influence inflammation and the immune system [4]. Altered activities of the kynurenine pathway (KP) enzymes and altered levels of the KP metabolites have been associated with cancer, autoimmune diseases, inflammation, neurologic diseases, and psychiatric disorders [3,5,6,7,8]. However, roles of the KP enzymes and metabolites in pathogenesis and progression of various diseases are a relatively less charted area of medical research. This review article discusses current findings and understanding of the alteration of the KP components in the central nervous system (CNS) and periphery, their possible contribution to pathogenesis and disease progression, and interventional strategies in main neurodegenerative diseases, searching for a risk, diagnostic, prognostic, predictive, and/or therapeutic biomarker that potentially ensures building a personalized therapy.

## 2. The Kynurenine Pathway and Kynurenines

The metabolism of the essential amino acid l-TRP takes place in two main pathways: the methoxyindole and the KP (Figure 1). The methoxyindole pathway is responsible for the synthesis of serotonin and melatonin. Only 1 to 5% of TRP is utilized for the methoxyindole pathway; thus, serotonin (5-HT) and melatonin synthesis depends on the availability of TRP which serves as a rate-limiting factor [9]. Approximately 95~99% of TRP is metabolized through the KP, which is responsible for the synthesis of bioactive molecules and nicotinamide adenine dinucleotide (NAD^+^). The efficient synthesis of NAD^+^ is crucial to maintain cell viability. NAD^+^ is an essential cofactor of the electron transport in the oxidative production of adenosine triphosphate (ATP) and hydride ion transfer in many enzyme reactions [10]. Disturbance of mitochondrial NAD^+^ impairs the respiratory chain and ATP synthesis resulting in energy depletion and cell death. NAD^+^ plays an important role in the brain’s glycogen storage, which is essential for CNS function [11] (Figure 1).

The cascade of KP produces several bioactive metabolites. The neurotoxic metabolites are 3-hydroxykynurenine (3-HK), 3-hydroxyanthranillic acid (3-HAA), and quinolinic acid (QUIN), while the neuroprotective ones are picolinic acid (PIC) and kynurenic acid (KYNA). 3-HK and 3-HAA generate reactive oxygen species (ROS) which raise the level of oxidative stress and lipid peroxidation [3]. QUIN is a weak, competitive agonist of the *N*-methyl-d-aspartate (NMDA) receptors containing the NR2A and NR2B subunits, being a neurotoxin, a gliotoxin, a proinflammatory mediator, and a pro-oxidant molecule [11]. QUIN stimulates synaptosomal glutamate release, inhibits glutamate uptake in the astrocytes, and participates in the generation of ROS which contributes to the depletion of endogenous antioxidants and the lipid peroxidation [12]. PIC is a neuroprotective metabolite that chelate a wide range of metals such as Cu, Fe, Ni, Zn, and Pb [13]. PIC exhibited immunomodulatory properties in vitro and showed antiviral, antifungal, antimicrobial, and antitumor activities in vivo and in vitro as well [13,14,15,16,17]. KYNA has antioxidant and anticonvulsant properties, scavenging ROS and suppressing inflammation. The reduced levels of KYNA may promote tissue damage and inflammatory cell proliferation in neurodegenerative diseases [18,19].

KYNA is a competitive antagonist of the ionotropic excitatory glutamate receptors including NMDA receptor, alpha (α)-amino-3-hydroxy-5-methyl-4-isoxazolepropionic acid (AMPA) receptor, and the kainate receptor [11]. KYNA binds to the strychnine insensitive glycine binding site of the NMDA receptors in lower concentration and to the glutamate-binding site of the NMDA receptors in higher concentration, respectively [12]. KYNA binds in a competitive or non-competitive manner to the strychnine insensitive glycine binding site, while it binds competitively to the glutamate-binding site resulting in the inhibition of overexcitation of the glutamatergic signal transduction [20]. NMDA receptors play a major role in synaptic transmission and neural plasticity of learning, memory, and other aspects of cognition [21,22,23,24]. At the AMPA receptor KYNA exhibits dual actions: KYNA stimulates the AMPA receptors in nanomolar to micromolar concentration, but competitively inhibits AMPA receptors in millimolar concentration [25,26]. KYNA inhibits the kainate receptors in higher concentrations in a range of 0.1–1 mM [27].

KYNA’s role in the α-7 nicotinic acetylcholine receptor (α7nAchR) remains controversial [21]. It was observed that KYNA inhibited the α7nAchR activity and increased non-α7nAchR expression in physiological and pathological conditions [28]. On the contrary, it was reported that KYNA did not modulate the α7nAchR activities in the hippocampus of the adult mouse brain [29]. KYNA activates the orphan G-protein receptor 35 (GPR35) expressed in glia and neurons of the hippocampus, which inhibits adenylate cyclase and depresses excitability and synaptic transmission in the hippocampus [30]. In addition, KYNA decreased body weight without negative influence on densitometry and mandibular bone [31].

Aryl hydrocarbon receptors (AHRs) are transcriptional factors that integrate environmental, toxic, dietary, and metabolic signals to regulate gene expression of enzymes such as cytochrome P450 [32]. AHRs were originally believed sensors of xenobiotic chemicals such as aryl hydrocarbon; however, AHRs are also activated by endogenous metabolites of the KP including KYN, KYNA, xanthurenic acid (XA), and cinnabarinic acid (CA), playing an important role in immunotolerance, autoimmunity, and tumorigenesis through the IDO/TDO-KYN-AHR feedback cycle [21]. In addition, AHRs control neuronal or glial migrations, neuronal development, differentiation, and synaptic function [21,33].

## 3. Kynurenines in the Brain, the Periphery, and the Gut-Brain Axis

### 3.1. Kynurenines in the Brain and the Periphery

In the brain l-TRP is transported through the highly selective semipermeable endothelial membrane blood-brain barrier (BBB) by the large neutral amino acid transporter [34]. TRP is converted into *N*-formyl-l-kynurenine by the rate-limiting enzyme indolamine-2,3-dioxygenase IDO. The *N*-formyl-l-kynurenine is converted to l-KYN by the formamidase. Approximately 40% of l-KYN is produced in the brain, while 60% of l-KYN is generated in the periphery, which can become available to the brain by being transported with the neutral amino acid carrier to cross the BBB [35].

l-KYN is metabolized through three different routes. The first route leads to the synthesis of KYNA by kynurenine aminotransferases (KATs). The second route leads to the synthesis of anthranilic acid (AA) by kynureninase and then 3-hydroxyanthranilic acid (3-HAA) by nonspecific hydroxylation. The third route leads to the synthesis of 3-hydroxykynurenine (3-HK) by kynurenine 3-monooxygenase (KMO), 3-hydroxyanthranilic acid (3-HAA), QUIN, and eventually NAD^+^. KATs also convert 3-HK to XA. PIC is produced from 3-HAA. The metabolites of the KP pathway are generally known as kynurenines. The first route takes place in astrocytes or skeletal muscle in the periphery, while the second and third routes are characteristic of microglial cells [3]. TRP, KYN, 3-HK are permeable, but KYNA and QUIN are impermeable to the BBB. Delivery of the BBB impermeable drugs with neuroprotective and antioxidant properties to CNS across the BBB is under extensive research [36,37].

In the periphery, several isoforms of the KP enzymes are expressed in different parts of the body. The indole ring of l-TRP is oxidized to *N*-formyl-KYN by tryptophan-2,3-dioxygenase (TDO), IDO-1, and IDO-2. TDO is expressed in the liver; IDO-1 throughout the body; and IDO-2 in the kidney, liver, and antigen-presenting cells. The major organ where the synthesis of KYNs takes place in the periphery is the liver. TDO is activated by the substrate TRP and stress hormone glucocorticoids, and ROS and inhibited by the KP metabolites 3-HK and NAD^+^, forming negative feedback loops [38].

IDOs are activated by lipopolysaccharides, pro-inflammatory cytokines including α-, beta, and gamma interferon (INF) and α tumor necrosis factor, and ROS and inhibited by antioxidant enzyme superoxide dismutase [39]. In addition, IDOs trigger immunosuppressive effects in T cells and myeloid-derived suppressor cells [40]. Thus, the KP metabolites play a key role in the communication between the nervous system, the immune system, inflammation, and redox homeostasis. Accordingly, it is not surprising that the disturbance of the KP is associated with neurologic diseases, autoimmune diseases, inflammatory diseases, and psychiatric disorders. A meta-analysis reported that an increased risk of depression was associated with inflammation in chronic illness through the KP [41]. A role of leptin in inflammation was discussed regarding various multifactorial diseases including neurologic diseases [42].

### 3.2. Gut-Brain Axis

Preclinical and clinical studies evidenced that the gastrointestinal microbiota influence the gastrointestinal (GI) physiology as well as the functions of the CNS by modulating different signaling pathways through the microbiota-gut-brain axis. The gastrointestinal microbiota is attributable to visceral pain, anxiety, depression, cognitive disturbance, and social behavior [43].

The healthy human microbiota shares a core microbiota composition and common trends from infancy to adulthood and old age. In the infant GI tract the main phyla are Bifidobacterium, Lactobacillus, Enterobacteriaceae and Staphylococcus. In adulthood the four dominant phyla are Bacteroidetes, Firmicutes, Verrucomicrobia, and Actinobacteria [44]. In the elderly the greater proportion of Bacteroides spp. with a distinct abundance patterns of Clostridium groups are the characteristic feature. The TRP metabolization is one of the most influenced and thus most important signaling pathways by the microbiota. l-TRP is one of essential amino acids, but it is readily biosynthesized by most plants and bacteria [45]. TRP transforms into several bioactive metabolites through the methoxyindole pathway and the KP, both of which influence the function of the GI nervous system and CNS with their changes of supply and availability.

But the microbiota not only influences the TRP source, but also synthetizes or degrades other neuroactive compounds too. For example, bacteria can biosynthesize QUIN, while unique prokaryotic enzymes can degrade KYNA [46,47]. KYNA is a NMDA receptor antagonist, being neuroprotective. On the contrary, QUIN is a NMDA receptor agonist, being neurotoxic [48]. The balance of QUIN and KYNA production and their function in the brain is known to be crucial, but the exact roles of these two metabolites in the GI tract are to be studied. KYNA has anti-inflammatory property into the GI tract, and both metabolites are involved in immunoregulation [49]. Therefore, there exists a complex system between the microbiota and the host. The diet and nutritional status of the host changes the microbiota composition which, furthermore, influences the metabolic pathways of the host, and vice versa. The modulation of the KP metabolism in the GI microbiota can be a new approach for the treatment of neurologic and psychiatric diseases. Furthermore, neurotoxic molecules may gain access to the CNS when the integrity of the BBB disrupts such as in inflammation. The influence of the gut microbiota on the brain function and behavior has become of emerging interest [11] (Figure 2).

## 4. Neurodegenerative Diseases

### 4.1. Alzheimer’s Disease

Alzheimer’s disease (AD) is the most common chronic neurodegenerative disease with an insidious onset of progressive cognitive deteriorations, particularly memory impairment. Motor or sensory dysfunctions are not prominent in the early stage. Motor and autonomic dysfunctions are associated with the comorbidities such as Parkinson’s disease (PD) with dementia, dementia with Lewy bodies, or vascular dementia [3]. Anxiety is common, besides apathy, depression, aggression, or sleep disorder [50]. Cortical atrophy of the frontal, temporal, and parietal lobes, enlargement of the temporal horn of the lateral ventricle, and atrophy of the entorhinal cortex, amygdala, and hippocampus are pathognomonic findings in patients with AD [51]. Abnormal deposit of insoluble proteinaceous material amyloid beta (Aβ) in the neuron and glial cells is mainly located in the atrophic lesions of AD patients [52]. Tau protein aggregates are associated with Aβ deposits, but it is considered secondary to amyloidosis [53]. Disturbance of calcium homeostasis was observed, and calcium-related proteins were proposed to be diagnostic and therapeutic biomarkers in AD [54].

The serum levels of TRP, KYNA, 3-HK, QUIN, and PIC were measured. The levels of 3-HK were significantly increased in the serum of patients with AD compared to those with major depression or with cognitive impairments. 3-HK is permeable to the BBB in contrast to a downstream metabolite QUIN and may be associated with higher levels of QUIN in the brain of AD patients [55]. An urgent need for biomarkers for the detection of the early stage AD was declared to expedite the early intervention by disease-modifying agents [55]. 

3-HK and other KP intermediates are possible candidates of early stage biomarkers [55]. The roles of the KP in the pathogenesis of AD were described [56,57,58,59,60]. QUIN was found localized with hyperphosphorylated tau in the cortical neurons of the brain of AD patients and to induce the phosphorylation of tau in human brains [57]. AD patients who have higher QUIN levels performed worse on the CAMCOG (the cognitive and self-contained part of the Cambridge Examination for Mental Disorders of the Elderly) test, suggesting the levels QUIN are associated with the cognitive impairment level [61] (Table 1, Table A1). QUIN is a strong oxidant, and the presence of oxidative stress was reported in AD, involving mitochondria dysfunction, microRNA, and microRNA-gene interaction [62,63].

Increased IDO-1 activity was associated with reduced cognitive performance, while IDO-1 inhibitor coptisine decreased the activation of microglia and astrocytes, prevented neuron loss, reduced Aβ plaque formation, and ameliorated impaired cognition in A b PP/PS1 mice [64,65]. KMO inhibitor JM6 prevented spatial memory deficits, anxiety related behavior, and synaptic loss in APP-Tg mice [66]. Furthermore, IDO is associated with the senile plaques [58]. Finally, the increased levels of KYNA were specific to cerebrospinal fluid (CSF) in AD, compared to that of frontotemporal dementia (FTD) and ALS [67] (Table 2, Table A1).

### 4.2. Parkinson’s Disease

PD is a progressive neurodegenerative disorder that predominantly affects motor functions including muscle rigidity, tremors, and changes in speech and gait. Main motor dysfunctions are bradykinesia, resting tremors, and rigidity which are largely due to the dopaminergic nigrostriatal denervation in the early stages of PD. However, psychobehavioral symptoms including psychosis, hallucinations, depression, and anxiety are not rare, which are present before the motor complaints [98,100]. Neurodegeneration and gliosis of the pars compacta of the substantia nigra (SNpc) and the presence of Lewy bodies (LBs) in pigment nuclei are pathognomonic of PD [101]. LBs contain the abnormal aggregates of misfolded alpha-synuclein (α-syn). Accumulation of aggregated α-syn in oligodendrocytes forms glial cytoplasmic inclusions. The mechanisms that govern α-syn fibrillization and LB formation in the brain remain poorly understood [102].

Alterations of TRP metabolism, glutamate excitotoxicity, and the gut-brain-axis have been shown associated with the pathogenesis of PD [44,103]. The identification of a risk marker is of particular interest because most of the dopaminergic neuros in the SNpc is not functioning at the time of the diagnosis [104]. KYNA levels and KYNA/KYN ratios were found significantly lower, while the levels of QUIN and ratios of QUIN/KYNA were observed significantly higher in the plasma of PD patients compared to healthy controls [75,105]. Those patients who were in advanced stage, Hoehn-Yahr stage more than 2, showed lower levels of KYNA and ratios of KYNA/KYN, while higher levels of QUIN and ratios of QUIN/KYNA compared to PD patients in early stage, Hoehn-Yahr stage ≤2 and healthy controls [105]. Moreover, receiver operating characteristic curve analysis suggested a QUIN/KYNA ratio as a potential biomarker for PD with good sensitivity and specificity. Stratified analysis showed that changes of the KYN pathway metabolites were more characteristic in PD patients in advanced stage [105]. Altered KYN metabolism and KYNA levels were reported in the brain samples of PD patients. The levels of KYNA were lower in the frontal cortex, putamen, and SNpc, while the levels of 3-HK were higher in the putamen, frontal cortex, SNpc, and CSF of PD patients [75,81]. The elevated 3-HK levels in CSF evidenced a possible excitotoxic disease mechanism in PD and 3-HK as a potential predictive biomarker [75] (Table 1, Table A2).

Single nucleotide polymorphisms (SNP) of IDO-1 rs7820268 and rs9657182 were found associated with the late onset of PD [106]. The activities of KAT II and levels of KYNA were increased in the red blood cells; however, the activities of KAT I and KAT II were lower and the levels of KYNA tended to be lower in the plasma of PD patients [92] (Table 2, Table A2). A systematic review reported the increased levels of neurotoxic KYNs and the decreased levels of neuroprotective KYNs in general, suggesting a significant shift toward the production of QUIN in the KP in PD [3]. Alteration of the KP is a distinguished characteristic in PD and may contribute to the pathogenesis of PD. Highly active retrotranposition competent LINE-1s was linked to the risk and progression of PD. making it a possible risk and therapeutic biomarkers [107]. Thus, the identification of PD-specific biomarkers in the blood, CSF, stool, or urine sample may make it possible to reveal the pathogenesis, make an early stage diagnosis, observe the disease progression, and monitor therapeutic effects.

### 4.3. Amyotrophic Lateral Sclerosis

Amyotrophic lateral sclerosis (ALS) is a group of progressive neurodegenerative disease which mainly affects neurons controlling voluntary muscles. ALS often presents fasciculation, myasthenia, or dysarthria initially. It involves the muscles responsible to move, speak, eat, and breathe in later stage [108]. ALS patients present a wide range of mild symptoms including autonomic, GI, cardiovascular, and neuropsychiatric manifestations including depression and anxiety [109,110,111]. The most common genetic mutations are the GGGGCC expansion in C9ORF72, present in approximately 30–47% of familial ALS cases and the SOD 1 mutations [112,113]. This hexanucleotide expansion is most often accompanied with the presence of cytoplasmic inclusions containing transactive response DNA-binding protein of 43 kDa (TDP-43) [114]. TDP-43 proteinopathy is characteristic to ALS [112]. TDP-43 is found in the lower motor neurons in the spinal cord and brainstem and the upper motor neurons in the motor cortex. In the late stage of ALS and ALS patients with dementia, TDP-43 can be found in the hippocampus, amygdala, and cortex [52]. New mutations were found in the genes of chromosome 9 open reading frame 72 (C9orf72), SOD 1, and senataxin in Hungarian ALS patients [115,116]. SNP of the vitamin D receptor gene rs7975232 (ApaI) was found associated with ALS [117]. Besides genetic predisposition, the pathogenesis of ALS is associated with ROS, mitochondrial dysfunction, intracellular calcium aggregation, and protein aggregation, glutamate excitotoxicity, and autoimmune inflammatory process [117,118,119,120]. 

The levels of KYNA was observed higher in the brain of patients with bulbar onset of ALS, compared to healthy control or patients with limb onset [87]. Moreover, the levels of KYNA were higher in CSF of patients with severe clinical status, compared to healthy controls [87]. Meanwhile lower levels of KYNA were detected in the serum of patients with severe clinical status, compared to healthy controls and patients with mild clinical status [87] (Table 1, Table A3). Therefore, the serum level of KYNA possibly indicates the severity of the disease and can be a potential prognostic biomarker. The neuronal and microglial expression of IDO were elevated and the levels of QUIN were higher in the motor cortex and spinal cord of ALS patients [87] (Table 2). The levels of TRP, KYN, and QUIN were elevated in CSF of ALS patients [87]. The levels of TRP, KYN, and QUIN were elevated, and the level of PIC was decreased in the serum of ALS patients [87]. Furthermore, the ratios of 3-HK/XA were decreased in the serum of ALS patients compared to patients with FTD [69] (Table 1, Table A3). The median survivaltime ranges from 20 to 48 months, but only 10–20% of ALS patients survive longer than ten years with worse prognosis in older age and bulbar onset. There is no option for the treatment of ALS. Thus, a search for predictive and therapeutic biomarkers are of particular interest. 

### 4.4. Huntington’s Disease

Huntington’s disease (HD) is an autosomal-dominant neurodegenerative disease with progressive and irreversible motor dysfunctions, leading to coordination problem, gait difficulties, cognitive dysfunction, and behavioral changes. Mild autonomic symptoms including orthostatic hypotension, excessive perspiration, and tachycardia are present in mild HD, while vegetative symptoms are prominent in the advanced stages [118]. Pathological findings in HD are degeneration and neural loss of the striatum, especially the caudate nuclei which innervate the cerebral cortex, pallidum, thalamus, brainstem, and cerebellum. The pathological changes correlate with disability. In the cerebellum, thalamus, and brain stem, abundant ballooned neurons were observed. Abnormal huntingtin proteins are associated with ballooning cell death which ruptures the membrane to swell like a balloon [118]. 

The activation of the neurotoxic branch of the KP is verified in the CNS. The levels of 3-HK and QUIN were elevated and the activity of 3-HAO was increased in the striatum where the loss of the nerve cell is the most prominent [76,98]. The levels of KYNA and the activity of KAT were decreased in the brain [88]. Toxoplasma gondii infection elevated the IDO activity in the brain and resulted significantly earlier death of the transgenic mouse model of HD compared to the HD mice without infection and the wild type, suggesting that the IDO activation accelerated the disease progression [119]. Lower TRP, higher KYN levels, and higher KYN/TRP ratios were observed in the serum of HD patients, suggesting the presence of higher IDO activity [70]. The levels of KYNA, the activity of KAT, and the levels of 3-HK and 3-HAA were all decreased in plasma [87]. The inflammatory status was well correlated with the levels of AA and the levels of TRP were negatively correlated with the severity of symptoms and the number of CAG repeats [71]. AA levels may be a good biomarker to indicate the inflammatory status in HD (Table 1, Table A4).

### 4.5. Multiple Sclerosis

Multiple sclerosis (MS) is an autoimmune demyelinating neurodegenerative disease. Common symptoms of MS range widely from motor dysfunction, autonomic symptoms to psychobehavioral manifestations including gait difficulties, paresthesia, vision problems, vertigo, incontinence, sexual problems, pain, cognitive dysfunctions, emotional changes, and depression [120]. Numerous glial scars, called plaques which develop in the white matter and spinal cord are pathognomonic to MS [121].

Disturbance of the KP and altered levels of KYN metabolites in MS patients were reported previously [77,91,122,123]. The levels of TRP were significantly lower in the serum and CSF samples of MS patients, suggesting the activation of the KP metabolism [72].The lower TRP level may be used as a potential biomarker in the screening of MS [71,73]. Proinflammatory cytokines including IFNs and TNF-α activated IDO-1 expression, resulting in the activation of the KP [124]. However, IDO-1 inhibition upon the disease induction significantly exacerbated the disease severity in the experimental autoimmune encephalitis (EAE) model of MS in mouse [125]. It was probably caused by the increased pool of available TRP which facilitated T cell proliferation. The activation and expression of IDO may become a useful biomarker to monitor the clinical course of relapsing-remitting multiple sclerosis (RRMS) and a predictive biomarker for the development of flares of MS. Moreover, therapeutic modulation of IDO activity may be beneficial in MS [91]. An imbalance of neurotoxic and neuroprotective KYN metabolites was considered involved in the pathogenesis of MS [7,77,92]. The activities of KAT I and KAT II enzymes were significantly higher in the red blood cells of MS patients compared to healthy control and the higher KAT activities were correlated with increased KYNA levels in the plasma of MS patients, suggesting the possible induction of neuroprotection against excitoneurotoxicity [94,95]. Furthermore, the levels of KYNA were increased in the CSF of MS patients during acute phase, while the KYNA levels were decreased in the inactive chronic phase of MS [77,78]. Monitoring the redox status including proteins, lipids, and nucleic acids together with the KP components was proposed to build a personalized treatment plan for MS patients [126] (Table 1, Table A5).

The activity and expression of KMO and the level of 3-HK were increased and KMO inhibitor Ro61- 8048 decreased the QUIN concentration in the spinal cord of EAE model of MS in rats [92]. A Ro61- 8048 prodrug KM6 significantly increased KYNA levels in mouse models of HD and AD [66]. The activity of KMO may be responsible for deviating from the KYNA branch towards the production of downstream neurotoxic metabolites. The activation of IDO-1 inhibits T cell activation, which appears beneficial to MS, but it can lead to the increased production of neurotoxic KYNs, eventually contributing to the progression of MS [66,127]. The KP metabolites were proposed to be potential prognostic and predictive biomarkers for MS. The levels KYNA and PIC were increased in RRMS, but not in secondary progressive multiple sclerosis (SPMS) or primary progressive (PPMS) and the levels of 3-HK and QUIN were increased in both SPMS and PPMS [116]. The QUIN/KYNA ratios were moderately correlated with the severity of MS [122] (Table 1, Table A5). Alteration of TRP metabolism is more relevant to the pathogenesis of MS than inflammation and a profile of the serum KP metabolites may be a suitable predictive biomarker for MS. Especially, the QUIN/KYNA ratio may become a useful predictive biomarker for neurodegeneration [122]. In general, the levels of KP metabolites can be suitable biomarkers for diagnosis of MS subtypes, monitoring the severity, and/or identifying therapeutic targets.

## 5. Other Relevant Diseases

Potential use of KYNs as biomarkers are under extensive research not only for neurologic diseases, but also psychiatric disorders. KYN and KYNA were found to be predictive biomarkers for the treatment of escitalopram in depression. KYNA is both a diagnostic and a predictive biomarker for depression as well [128]. 3-HK and KYNA were proposed to be prognostic biomarkers of depression and disability in poststroke patients [129]. Other KP metabolites such as XA and CA were rarely documented. An extremely low concentration of CA was reported to have anti-psychotic activities in mice and the levels of CA was reduced in the prefrontal cortex in patients with schizophrenia [130]. The glutamatergic nervous system was proposed to be a therapeutic biomarker for mood disorders including depression [131]. Furthermore, a simultaneous intervention in the NMDA receptor and α7nAchR was suggested by novel combination for the treatment of schizophrenia [132]. Longitudinal plasma samples were studied in search of a certain plasma protein profiles as a predictive biomarker for the treatment of depression [133]. Therapeutic biomarkers are under rigorous search for depression, anxiety, and dementia through endogenous neuropeptides, agonists, and their synthetic analogues [134,135,136,137]. Omega-3 polyunsaturated fatty acids which bind G protein-coupled receptor GPR120 in the GPR120 signaling pathway was proposed to be a therapeutic biomarker for the treatment of schizophrenia [138]. The treatment of metabolic dysfunction by nutraceuticals in ageing and neurodegenerative diseases was proposed [139]. Biomarkers are not only limited to molecules, but can also be any measurable indicators for risk, diagnosis, prognosis, disease course, and therapeutic targets. Depression was reported a risk factor for AD and dementia, and dyslipidemia treatment reduced this risk in patients with diabetes mellitus. Thus, depression is a risk biomarker and preventable in patients with dyslipidemia [140]. The presence of depression after acute stroke and transient ischemic attack increased mortality and disability within 12 months, suggesting depression as a prognostic biomarker in cerebral ischemia [141]. Depression and anxiety can be treatable by psychedelic psilocybin in patients with terminal illness [142]. Interestingly, depression is a single psychobehavioral component of dementia, which can be ameliorated by animal-assisted and pet-robot interventions in dementia patients [143]. Depression is indeed a therapeutic biomarker.

## 6. Conclusions and Future Perspective

The lack of appropriate biomarkers to make a diagnosis and follow-up therapy is seriously hampering the application of personalized medicine to neurodegenerative diseases. Consensus on the methodologies and validations are missing in many cases. Consequently, thousands of possible biomarkers were documented in the literature, but only hundreds are in clinical use. It is important to study more populations and repeat the analysis in different cohorts. After verification, a potential biomarker must be tested in the other population worldwide including Caucasians, Asians, and Africans. Furthermore, non-invasive samples such as sweat, tear, urine, and stool are to be explored for biomarker research, in addition to the peripheral sampling.

Profiling metabolomic data may contribute to revealing the state of metabolism. The roles of the microbiota in the gastrointestinal-brain axis are to be explored to profile the colony of the microbiota in neurodegenerative diseases. Magnetic resonance imaging with imaging biomarkers may be able to assess the status of the BBB integrity to estimate the influence of the gastrointestinal microbiota on the CNS in the microbiota-gut-brain axis. Therefore, the KP profiles, metabolomic profiles, the gastrointestinal microbiota colony profiles, the BBB integrity index may all serve to integrate into a battery of powerful biomarkers to expedite building a personalized treatment plan for neurologic diseases beyond neurodegenerative diseases discussed in this article, such as strokes and migraine as well as psychiatric disorders such as depression, anxiety, and schizophrenia. 

## Figures and Tables

**Figure 1 ijms-21-09338-f001:**
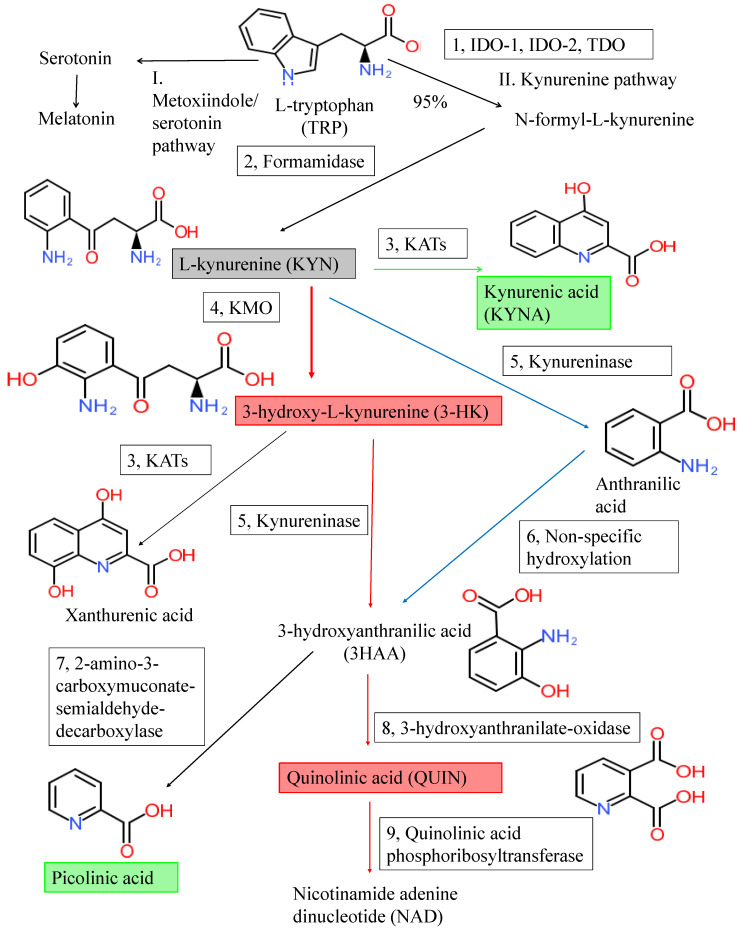
Tryptophan metabolism and the kynurenine pathway. The kynurenine pathway (KP) is the main degradation route of tryptophan (TRP) metabolism producing the end-product nicotinamide adenine dinucleotide (NAD). The indoleamine-2,3-dioxygenase-1 and 2 (IDO-1 and IDO-2), and the tryptophan 2,3-dioxygenase (TDO) (1) are the first rate-liming enzymes that convert the l-TRP to *N*-formyl-l-kynurenine. *N*-formyl-l-kynurenine is converted by formamidase (2) to l-kynurenine (l-KYN) (gray box). l-KYN is metabolized into various bioactive compounds: the neuroprotective metabolites are kynurenic acid and picolinic acid (green boxes), while the neurotoxic ones are 3-hydroxy-l-kynurenine (3-HK) and quinolinic acid (red boxes). The main enzymes of the KP are following: 1: tryptophan 2,3-dioxygenase (TDO) and indoleamine 2,3-dioxygenase 1 and 2 (IDO-1 and IDO-2), 2: formamidase, 3: kynurenine aminotransferases (KATs), 4: kynurenine-3-monooxygenase (KMO), 5: kynureninase, 6: non-specific hydroxylation, 7: 2-amino-3-carboxy-muconate-semialdehyde decarboxylase, 8: 3-hydroxyanthranilate oxidase, 9: quinolinic acid phosphoribosyltransferase.

**Figure 2 ijms-21-09338-f002:**
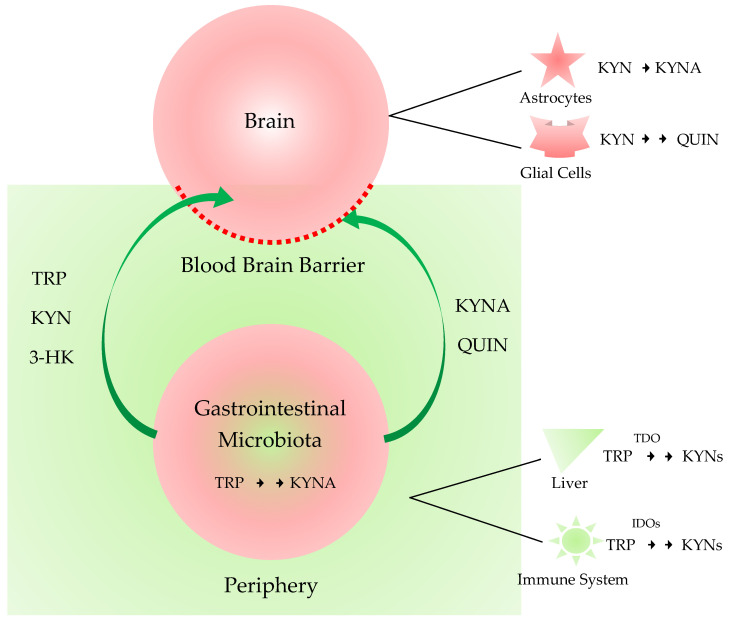
The central nervous system and periphery sequestrated by the blood-brain-barrier and the microbiota-gut-brain axis. The impermeable peripheral metabolites are sequestrated from the central nervous system (CNS) by the blood-brain barrier (BBB). The gastrointestinal microbiota changes the composites of the peripheral metabolites. Tryptophan (TRP), kynurenine (KYN), 3-hydroxykynurenine (3-HK) are permeable, but kynurenic acid (KYNA) and quinolinic acid (QUIN) are impermeable to the BBB. When the integrity of the BBB is compromised such as in inflammation, the microbiota influences the brain function and thus behavior through the microbiota-gut-brain axis. TDO: tryptophan 2,3-dioxygenas; IDO: indoleamine 2,3-dioxygenase.

**Table 1 ijms-21-09338-t001:** Changes of metabolites of the kynurenine pathway in neurologic diseases. ↑: increase, ↓: decrease, *: tendency, not statistically significant.

Metabolites	Alzheimer’s Disease		Parkinson’s Disease		Amyotrophic Lateral Sclerosis		Huntington’s Disease		Multiple Sclerosis	
	CNS	Peripheral	CNS	Peripheral	CNS	Peripheral	CNS	Peripheral	CNS	Peripheral
TRP	-	↓ [61] ↓ [68]	-	-	↑ [69] ↑ [69]	↑ [69]	-	↓ [70] ↓ [71]	↓ [72]	↓ [72] ↓ [73]
QUIN	↑ [56] ↑ [59] ↑ [60]	↓ [61] ↑ [68]	↑ [74]	↑ [75]	↑ [69]	↑ [69]	↑ [76]	-	↑ [77]	↑ [78]
3-HK	-	↑ [55] ↑* [68] ↑ [79]	↑ [80] ↑ [81]	↑ [74]	-	-	↑ [76] ↑ [82]	↓ [83]	↑ [77]	↑ [78]
KYNA	↑ [60] ↑ [67] ↓ [84] ↑ [85]	↓ [68] ↓ [79]	↓ [74] ↓ [80]	↓ [75] ↓ [86]	↑ [87]	-	↓ [88] ↓ [89] ↓ [90]	-	↓ [91] ↑ [92]	↑ [78] ↑ [93]
AA	-	↑ [66] ↑* [68]	-	-	-	-	-	-	-	-
KYN	-	↑* [68] ↑ [79]	-	↑ [94]	↑ [85]	↑ [85]	-	↑ [67]	-	-
XA	-	↓ [61]	-	-	-	-	-	-	-	-
3-HAA	-	↓ [61]	-	↓ [74]	-	-	-	↓ [83]	-	-
PIC	-	-	-	-	-	↓ [69]	-	-	-	↑ [78]

**Table 2 ijms-21-09338-t002:** Changes of metabolites of the kynurenine pathway in neurologic diseases. ↑: increase, ↓: decrease.

Enzyme Activity	Alzheimer’s Disease		Parkinson’s Disease		Amyotrophic Lateral Sclerosis		Huntington’s Disease		Multiple Sclerosis	
	CNS	Peripheral	CNS	Peripheral	CNS	Peripheral	CNS	Peripheral	CNS	Peripheral
IDO	↑ [56] ↑ [59]	↑ [64] ↑ [68]	↑ [80] ↑ [95]	↓ [56] ↑ [86]	↑ [69]	-	-	↑ [70] ↑ [71] ↑ [83]	-	↑ [91]
TDO	↑ [59]	-	-	-	-	-	-	-	-	-
KAT I	↑ [85]	-	↓ [96] ↓ [97]	↓ [97]	-	-	↓ [88] ↓ [90]	↓ [83]	-	↑ [93]
KAT II	↑ [98]	-	↓ [97] ↑ [97]	-	-	-	↓ [88] ↓ [90]	↓ [83]	-	↑ [93]
KMO	-	-	-	-	-	-	-	-	↑ [92]	-
3-HAO	-	-	-	-	-	-	↑ [99]	-	-	-

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
