# Peer review of "Searching for Peripheral Biomarkers in Neurodegenerative Diseases: The Tryptophan-Kynurenine Metabolic Pathway"

_ijms, 2020, doi:10.3390/ijms21249338_

Round 1

Reviewer 1 Report

Following the analysis of the manuscript titled "Searching for Peripheral Biomarkers in Neurodegenerative Diseases: The Tryptophan Kynurenine Metabolic Pathway", I would like to express my appreciation towards the authors as the article's topic is very interesting. The manuscript is well documented and the presentation of the information is clear and properly structured.

Author Response

Dear Reviewer, 

Thank you for your kind support. We sincerely appreciate your critical reading and evaluation of our manuscript.

Best regards,

Authors

Reviewer 2 Report

The paper “Searching for Peripheral Biomarkers in Neurodegenerative Diseases: The Tryptophan-Kynurenine Metabolic Pathway” describes the possible use of kynurenines (KYNs) and the kynurenine pathway (KP) enzymes as potential biomarkers for neurodegenerative disorders. I consider this review provides interesting data, as little is known on the role of these metabolites in neurodegenerative diseases and lesser about their use as potential biomarkers. Even though the paper is suitable for publication, some points could be improved:

Minor points:

  • How are these metabolites quantified? Could they be included in clinical routine analysis? Which biofluid is used for measurement: plasma, serum? A brief explanation on the clinical analysis would provide a better picture of the validity of these metabolites as biomarkers for neurodegenerative diseases.
  • Figure 1 would be improved if the initials of the enzymes catalyzing the reactions would be included (next to the numbers or without the numbers), because it is hard for the reader to follow-up later in the text.
  • Figure 2 could be improved. It is not clear where are these metabolites produced and the role of the gastrointestinal microbiota. A representation of where in the cell (and which type of cells if it is specific) are these metabolites produced.
  • Section 2 describes the role of these metabolites, especially KYNA. First the role of QUIN is explained, then KYNA, then AHRs and then KYNA. Sometimes it is hard for the reader to follow. Could the information in the section be organized in a better way?
  • Section 3 “Kynurenines in the Brain, the Periphery and the Gut-Brain Axis”, would be improved if it included subheaders as in section 4.
  • In the conclusions section the authors describe mainly advances regarding psychiatric disorders that (1) haven´t been discussed previously in the text and (2) are not necessarily neurodegenerative disorders. I would include another section about psychiatric disorders if the authors consider this information as relevant (lines 370-394) and keep as conclusions the information in lines 394-411.

English grammar and spelling should be revised along the text. Some examples are:

  • Line 47 “influence” instead of “influence”.
  • Lines 60 and 62: TRP instead of TRY. If it is not a spelling mistake, explain.
  • Line 312: “proteins are associated” instead of “proteins is associated”.
  • Line 322: “decreased in plasma” instead of “decreased in the plasma”.
  • Line 339: “mice” instead of “mouse”.
  • Line 369: “Conclusions” instead of “Conclusion”.
  • Line 372: “were” instead of “was”.

Author Response

Dear Reviewer,

We sincerely appreciate your critical reading and valuable suggestions.

How are these metabolites quantified? Could they be included in clinical routine analysis? Which biofluid is used for measurement: plasma, serum? A brief explanation on the clinical analysis would provide a better picture of the validity of these metabolites as biomarkers for neurodegenerative diseases.

Thank you for your valuable comments. We included detailed information in the supplementary tables. In the main text, we presented simplified versions to help readers to grasp the data and make them more informative. The supplement presents more detailed information including sample types. In general, most of the samples were the plasma, serum, and cerebrospinal fluid, while few samples were the urine and red blood cells. Unfortunately, non-invasive samples including sweat, tear, urine, and stool are underrepresented in biomarker research of neurodegenerative diseases. Metabolites were quantified in most of the cases by high performance liquid chromatography or gas chromatography coupled with tandem mass spectrometry. The analytical techniques are expected to be a part of routine laboratory tests in the future for personalized medicine.

Figure 1 would be improved if the initials of the enzymes catalyzing the reactions would be included (next to the numbers or without the numbers), because it is hard for the reader to follow-up later in the text.

We corrected the figure accordingly, in addition to making it more visual. We appreciate your valuable suggestion.

Figure 2 could be improved. It is not clear where are these metabolites produced and the role of the gastrointestinal microbiota. A representation of where in the cell (and which type of cells if it is specific) are these metabolites produced.

Thank you for precious suggestion. We added more information on the brain and periphery but tried to make it simple to remain informative to readers.

Section 2 describes the role of these metabolites, especially KYNA. First the role of QUIN is explained, then KYNA, then AHRs and then KYNA. Sometimes it is hard for the reader to follow. Could the information in the section be organized in a better way?

We corrected the order. The metabolites were described in the section to follow the logical sequence. We sincerely appreciate your thoughtful recommendation.

Section 3 “Kynurenines in the Brain, the Periphery and the Gut-Brain Axis”, would be improved if it included subheaders as in section 4.

Thank you for your kind suggestions. We added subheaders in the section.

In the conclusions section the authors describe mainly advances regarding psychiatric disorders that (1) haven´t been discussed previously in the text and (2) are not necessarily neurodegenerative disorders. I would include another section about psychiatric disorders if the authors consider this information as relevant (lines 370-394) and keep as conclusions the information in lines 394-411.

We made a subsection for the other relevant diseases before the conclusions. We sincerely appreciate your thoughtful suggestions.

English grammar and spelling should be revised along the text. Some examples are:

  • Line 47 “influence” instead of “influence”.
  • Lines 60 and 62: TRP instead of TRY. If it is not a spelling mistake, explain.
  • Line 312: “proteins are associated” instead of “proteins is associated”.
  • Line 322: “decreased in plasma” instead of “decreased in the plasma”.
  • Line 339: “mice” instead of “mouse”.
  • Line 369: “Conclusions” instead of “Conclusion”.
  • Line 372: “were” instead of “was”.
  • Line 372: “were” instead of “was”.

Thank you for your kind help. We corrected the grammars and typos mentioned above. Furthermore, the manuscript was proofread by Fanni Tóth, PhD. and Zsuzsanna Fülöpné Bohár, PhD.

Best regards,

Authors